# Lifestyle as a Modulator of the Effects on Fitness of an Integrated Neuromuscular Training in Primary Education

**DOI:** 10.3390/jfmk9030117

**Published:** 2024-07-02

**Authors:** Blanca Roman-Viñas, Fidanka Vasileva, Raquel Font-Lladó, Susana Aznar-Laín, Fabio Jiménez-Zazo, Abel Lopez-Bermejo, Victor López-Ros, Anna Prats-Puig

**Affiliations:** 1University School of Health and Sport (EUSES), University of Girona, 17190 Girona, Spain; fvasileva@idibgi.org (F.V.); rfont@euses.cat (R.F.-L.); victor.lopez@udg.edu (V.L.-R.); aprats@euses.cat (A.P.-P.); 2Department of Physical Activity and Sport Sciences, Blanquerna, Universitat Ramon Llull, 08022 Barcelona, Spain; 3Biomedical Research Centre in Physiopathology of Obesity and Nutrition (CIBERobn), Instituto de Salud Carlos III (ISCIII), 28029 Madrid, Spain; 4Pediatric Endocrinology Research Group, Girona Institute for Biomedical Research, 17003 Girona, Spain; 5Specific Didactics Department (Serra Húnter Fellow), University of Girona, 17003 Girona, Spain; 6Chair of Sports and Physical Education & Spanish Olimpic Committee, Universitat de Girona, 17003 Girona, Spain; 7PAFS Research Group, Faculty of Sports Sciences, University of Castilla-La Mancha, 45071 Toledo, Spain; susana.aznar@uclm.es (S.A.-L.); fabio.jimenez@uclm.es (F.J.-Z.); 8Pediatric Endocrinology Group, Girona Biomedical Research Institute, 17190 Girona, Spain; alopezbermejo@idibgi.org; 9Department of Pediatrics, Dr. Josep Trueta Hospital, 17007 Girona, Spain; 10Department of Medical Sciences, University of Girona, 17003 Girona, Spain; 11Faculty of Education and Psychology, University of Girona, 17003 Girona, Spain; 12Research Group of Clinical Anatomy, Embryology and Neuroscience (NEOMA), Department of Medical Sciences, University of Girona, 17003 Girona, Spain

**Keywords:** physical education, fitness, children, lifestyle habits, parental education

## Abstract

The objective was to evaluate changes in fitness after an integrated neuromuscular training (INT) intervention in primary school children and to evaluate how lifestyle behaviors and parental education modulate these changes. One hundred and seventy children (7.45 ± 0.34 years; 52% girls) were included. Cardiorespiratory fitness (half-mile run test), a 10 × 5 m shuttle run test, standing broad jump (SBJ), handgrip dynamometer, body mass index (BMI) and fat mass percentage (FM%) were assessed before and after the 3-month intervention (20 min of INT in the physical education class, twice per week). The Mediterranean diet (MD), sleep time and parental education level (PEL) were evaluated by questionnaires, and adherence to physical activity (PA) recommendations was measured with a triaxial accelerometer before the intervention. After the intervention, there were improvements in the 10 × 5 test and the SBJ. Only girls had improvements in the handgrip test, BMI SDS and FM%. After correcting for confounding variables, only BMI was significantly improved whereas strength improved in the participants non-compliant with the PA recommendations or pertaining to families of high PEL. The INT produced improvements in fitness in a brief period and in different subgroups of pupils (inactive and with diverse sociocultural environments).

## 1. Introduction

Physical fitness (PF) is a significant marker of health at an early age [1], increases the likelihood of being active as an adult [2] and reduces the risk of having several cardiovascular risk factors [3,4]. There exists evidence showing declining levels of aerobic capacity, muscular strength, flexibility and speed in children in the last four decades across the world [5]. In addition to physical activity (PA) [6], diet and sleep duration [7], genetics [8] and motor competence (MC) [9] have been correlated with having adequate levels of PF during childhood. According to the conceptual model of Stodden [9], there is a dynamic relationship between PF, MC, perceived motor competence and PA from early childhood to young adulthood. PA alone is not enough to achieve adequate levels of MC [10] and fitness, and therefore children must be taught to acquire and develop fundamental motor skills [11] that increase the likelihood of being physically active and reduce sedentary time later in life [6]. Cross-sectional and longitudinal studies conducted after the Stodden model was published have provided some evidence that learning certain motor skills at early ages can improve some domains of physical activity [12]. PF and MC are skills that share common neuromuscular mechanisms, whereas PA is a behavioral attribute that coexists with others such as a healthy diet and good sleep habits in children with a favorable weight status [13] and good levels of fitness [14]. The adoption and maintenance of healthy lifestyles requires high self-determined motivation [15] which, applied at the physical activity level, suggests that the improvements in autonomous motivation for activities conducted in school will improve motivation for activities conducted outside of school [16].

Given the complex interaction between PA, MC and fitness and their relationship with health markers, there is a need to ensure the acquisition and development of all those behaviors and skills at an early age to establish the basis of a healthier life. School time is an opportunity for children to be physically active, by using active transportation, by being active at recess time and obviously, in physical education (PE) class. In addition, compulsory education ensures that every child can learn and thrive regardless of their background or socioeconomic status. PE is included in the school curricula in all of the European countries, although the time allocated to the classes and their quality vary significantly. According to UNESCO [17], a good quality PE curriculum should provide children with learning experiences that “… help them acquire the psychomotor skills, cognitive understanding, and social and emotional skills they need to lead a physically active life.” Unfortunately, the time children are active during PE classes is quite low [18]. The barriers for this situation include a lack of resources or skills to adapt the pedagogical methodology to the diversity of pupils, a lack of time or children who are not motivated in PE classes because they are based on traditional sports and promote competitiveness [17]. In terms of increasing physical activity outside of school, certain interventions adopting an autonomy-supportive teaching style in PE class to improve children’s affective and emotional perception have shown increases in leisure time PA [19]. Regarding PF and MC, some reviews exploring the effect of PE interventions have shown effects of different magnitude on body mass index (BMI), fundamental motor skills, cardiorespiratory fitness and muscular strength depending on the type and objectives and duration of the interventions, the age group and the gender of the participants [3,20]. Some of the interventions evaluated in those reviews consisted of the adoption of innovative learning theories to help teachers and motivate pupils to progress in the acquisition of those skills needed to improve PA. Integrative neuromuscular training (INT) is designed to develop and improve fundamental motor skills and fitness [21]. Some studies have shown that incorporating such a methodology into the PE class can improve MC and some fitness components in school-age children [22,23,24].

Therefore, the main objective of this study was to evaluate the effects of a 3-month INT intervention added into the PE class on PF (speed-agility, cardiorespiratory fitness, lower and upper limbs’ muscular strength) and anthropometric variables (BMI and fat mass) in 7–8-year-old children, according to sex and lifestyle habits (PA, Mediterranean diet adherence and sleep time). Furthermore, as parental education is a common correlate of MC [25], PA [26] and fitness [27], a secondary objective was to study how parental educational influences the effects of the PE intervention on PF and anthropometric variables.

## 2. Materials and Methods

### 2.1. Subjects

The data were obtained from the Physical Education, Health and Children (PEHC) study, a cluster randomized controlled trial to evaluate the effects of an INT program incorporated into PE class to improve MC in children in the second year of primary school [23].

Five schools in the northeast of Spain accepted to participate in the study. The school principals were informed about the aim and procedures of the study and their approval to participate was obtained. The inclusion criteria to participate were being in the second year of primary education (aged 7 to 8 years) and provide a parental signed informed consent. The exclusion criteria were (a) major congenital abnormalities, (b) evidence of chronic illness or chronic medication use, (c) musculoskeletal or neurological disease, (d) functional limitations and (e) pain or dysfunction in the upper or lower extremities during exercise. In addition, children not delivering the requested questionnaires, not performing the MC and PF tests or not attending at least 80% of the PE classes were excluded from the analysis. After attending an explanatory meeting delivered by the research team, parents/legal guardians accepting to participate were provided with a questionnaire with information about socioeconomic data, ethnic origin, and food habits (KIDMED questionnaire) and a diary to register the accelerometer usage.

The study was conducted according to the Declaration of Helsinki ethical guidelines and was approved by the Institutional Review Board of the Dr. Josep Trueta Hospital.

Schools were assigned to the intervention group (3 schools) or the control group (2 schools) by means of a specific software program. Once the study was completed, the schools that acted as controls were offered to receive the INT program.

The INT program was carried out during the 20 min warm up in the PE class, 2 days per week for a total of 24 sessions, and included 3 units focused on 3 fundamental motor skills: locomotor, stability and object motor skills [23]. Each unit included 8 training sessions, 3 based on structured gamified sessions, focused firstly on the body control and awareness and then on FMS and physical conditioning components and 5 sessions with circuits of 7 tasks (each subject completed 12 repetitions of each circuit with 1 min of rest between each). In addition, the 5 circuit sessions of each unit were made up of 5 levels of complexity, which included the main components of physical conditioning that characterize INT (dynamic stability, coordination, strength, plyometrics, speed and agility and fatigue resistance). All sessions were supervised by 2 professionals: (a) the PE teacher previously trained in the INT program in 4 sessions: familiarization with the INT material (1 h), theoretical framework (1 h), example of the INT program in practice (2 h) and how to give feedback (2 h); and (b) an INT expert (Sports Scientist) different to the observer and statistics analyzer to mentor the PE teacher to improve the quality of instructions and feedback. After the 20 min INT program, children continued with the regular PE class. The control group performed the regular 60 min PE class.

The study was developed during the 2016–2017 and 2017–2018 academic years. The baseline assessment was conducted the week before the study was initiated, by trained researchers during school hours in the school facilities. The same researchers conducted the final evaluation one week after the study was finalized. The researchers conducting the measurements were not blinded.

### 2.2. Measurements

Anthropometric variables were measured in children wearing light clothes. Weight and fat mass percentage (FM%) was measured with a calibrated scale (Portable TANITA, 240MA, Amsterdam, The Netherlands) and height was measured using a wall-mounted stadiometer (SECA SE206, Hamburg, Germany). Body mass index (BMI) was calculated as weight divided by the square of height in meters. BMI z-scores were standardized according to age- and sex-adjusted values from regional normative data [28].

Speed-agility was assessed by means of a 10 × 5 m shuttle run test from a standing start [29]. The total time needed to complete five cycles (back and forth) between two lines separated by five meters was recorded in seconds. Both feet had to cross the lines each time to be valid. Cardiorespiratory fitness (CRF) was assessed by means of a half-mile run test. The aim of this test was to complete the half-mile run, around two cones separated by 40 m, in the quickest possible time [30]. The total time taken to run half a mile was recorded in minutes. Lower-limb muscular strength (LLMS) was assessed using the standing broad jump (SBJ) [31]. Children jumped as far as possible with their feet together while remaining upright. The longest distance (measured in cm) of two attempts was used in the analysis. Upper-limb muscular strength (ULMS) was measured in kg using an analog hand dynamometer (TKK 5001, Grip-A, Takei, Tokyo, Japan). The precision of the dynamometer is 0.1 kg. Children were asked to squeeze the dynamometer gradually and continuously for at least 5 s. The test was performed twice for each hand. The mean value of both hands was used for the analysis.

PA was measured with the triaxial Actigraph GT3X accelerometer (Actigraph, Pensacola, FL, USA) validated for their use in children [32]. Children were instructed to wear the accelerometer at the hip, 24 h a day during seven days, and to remove it only for water-related activities. A calendar was used to register the hour they took off the accelerometer and the moment it was located again. Parents used the same calendar to indicate the hour their children went to bed and woke up. The devices were delivered and collected every week at the school. They were programmed to begin data collection at midnight after being delivered to the participants. The Actilife software (version 6.13.4.0, ActiGraph LLC, Pensacola, FL, USA) was used to initialize, download and analyze the data. The accelerometers were programmed to register data at a sampling rate of 100 Hz, with an epoch length of 15 sec and with a normal filter. Once collected, data were reintegrated to 60 s epochs. For the present analysis, we excluded the data collected while sleeping. Periods of 20 min of consecutive zero counts were selected as non-wear time and removed from the analysis. Only those registers with at least four days (with at least one weekend day) of 10 h of valid wear time were included for the analysis. The time spent in different PA intensities was defined applying the Evenson cut-off points [33]: sedentary (0 to 100 counts per minute (CPM)), light intensity (101–2295 CPM), moderate intensity (2296–4011 CPM) and vigorous intensity (4012 or more CPM). Those children with an average time of moderate-to-vigorous PA of 60 min or more over all valid days of measurement were classified as compliant with PA recommendations [34].

The Mediterranean diet is characterized by the intake of a great number of vegetables, fruits, bread and other forms of cereal, rice, beans and nuts. It includes virgin olive oil as the principal source of fat, moderate amounts of dairy products (basically cheese and yogurt), moderate amounts of fish, red meat in low amounts and wine consumed in small quantities (for adults) [35]. Mediterranean diet (MD) adherence was evaluated with the KIDMED test, a 16-item validated questionnaire to measure adherence to the MD in Spanish children [36]. Answers showing positive adherence to the MD receive one point, and those showing a negative adherence receive minus one. The global score ranges from 0 to 12 points. The variable was dichotomized as non-compliant (less than 8 points) and compliant (8 points or more) with the MD pattern. The parents or legal guardians answered the questionnaire.

Sleep time was obtained from the calendars administered to the families. The cut-off point to define appropriate sleep duration was 10 h.

To evaluate the parental educational level (PEL), parents/guardians were questioned about their level of studies and the highest level achieved by either of them was considered for the analysis. The variable was dichotomized as low (no formal education, primary or secondary education) and high educational levels (higher than compulsory education or university degree).

### 2.3. Data Analysis

Program package SPSS version 22.0 (SPSS Inc., Chicago, IL, USA) was used to perform statistical analysis. All variables were tested for normality using the Kolmogorov–Smirnov test. Descriptive data are presented as the mean ± standard deviation (SD). Percentages are also used to present some variables (adherence to lifestyle recommendations). Differences between groups (boys/girls and control/intervention) were analyzed by an unpaired *t*-test, while differences between pre- and post-intervention were assessed using the univariate general linear model. We further explored outcome differences by gender, adherence to lifestyle recommendations (PA, MD and sleep time) and PEL. Logistic regression models were used to calculate odd ratios (OR’s) and 95% confidence intervals (CI) to determine differences between PF and anthropometric variables (equal and above or below percentile 50) according to the intervention in all children and in groups of PA adherence and PEL. Significance was set at *p* < 0.05.

## 3. Results

From the 281 children attending the five schools, 204 met the inclusion criteria and delivered written informed consent (105 children from the schools assigned to the intervention and 99 from the control schools). The final analysis included 170 children, 74 in the control group and 99 in the intervention group (Figure 1).

Girls (Table 1) showed higher percentages of FM% than boys (*p* = 0.001), while males outperformed females in PF (all *p* < 0.0001) except for the ULMS, with no differences between genders. Boys spent more time (around 15 min; *p* < 0.0001) in moderate and vigorous PA and showed a higher adherence to the PA recommendations than girls (*p* < 0.0001). A total of 31.2% of the sample showed a good adherence to the MD recommendations, with no differences between genders. Compared to boys, girls showed a higher percentage of participants adhering to the three lifestyle factors (5.7%) but also a higher number of participants not following either of them (26.2%).

Table 2 shows the effect of the intervention by gender. After the intervention, the time to complete the 10x5 test was reduced in boys (Cohen’s d = 0.70; *p* < 0.0001) and girls (Cohen’s d = 0.51; *p* < 0.0001). The distance in the SBJ test increased in boys (Cohen’s d = 0.32; *p* = 0.012) and girls (Cohen’s d = 0.51; *p* = 0.004). Only girls obtained significant improvements in the handgrip test, BMI SDS and FM%.

Table 3 shows the effect of the intervention in all the samples and by adherence to the PA recommendations and PEL. Both the non-compliant and compliant groups improved their speed-agility (−1.24 ± 1.19 s (Cohen’s d = 0.53) and −1.84 ± 1.35 s (Cohen’s d = 0.72), respectively; both *p* < 0.0001), LLMS (5.64 ± 7.67 cm (Cohen’s d = 0.37; *p* = 0.001) and 5.44 ± 11.12 cm (Cohen’s d = 0.32; *p* = 0.02), respectively) and ULMS (0.61 ± 1.28 kg (d = 0.27; *p* = 0.008) and 0.48 ± 1.06 kg (d = 0.22; *p* = 0.003), respectively). The intervention produced higher improvements in speed-agility and lower- and upper-limb muscle strength compared to the improvements in the control group in participants who were not compliant with the PA recommendations. Interestingly, improvements in anthropometric variables such as BMI SDS and FM% were only seen in children in the intervention group and non-compliant with PA recommendations (*p* = 0.025 and *p* = 0.015, respectively). Referring to PEL, children from families with a low PEL in the intervention group improved their speed-agility compared to controls significantly (*p* = 0.006) and with a higher effect size. In addition, pre–post comparisons in both interventions and control groups for LLMS, and in the intervention group only for ULMS, showed statistically significant differences between the two measurement points. Children from families with a high PEL in the intervention group improved their speed-agility (*p* < 0.0001), CRF (*p* = 0.006) and ULMS (*p* = 0.003). Regarding anthropometric variables, the intervention produced a statistically significant reduction in BMI and FM% in children whose parents had a low PEL. There was a reduction in BMI in the control group and in FM% in the intervention group in children pertaining to families with a high PEL.

Table 4 shows the results of the logistic regression analysis (OR and CI) considering the differences in PF and anthropometric variables according to the intervention in all children and by the level of compliance to PA recommendations and by PEL. The intervention appeared to have a protective factor that could avoid the increases in BMI, showing positive effects in all children and especially in children non-compliant with PA recommendations and in children in the high PEL category. Finally, the INT intervention positively affects the increase in strength in children who follow the PA recommendations and in those in the high PEL category.

## 4. Discussion

The present study shows improvements in speed-agility and muscular strength in both boys and girls after a three-month intervention incorporating 20 min of an INT program as a warmup in PE classes. Interestingly, girls and participants not being sufficiently physically active also benefit from improvements in BMI SDS and FM%. For speed-agility and muscle strength, those children not adhering to PA recommendations at baseline found similar benefits as those being compliant. The intervention produced a higher effect in speed agility and ULMS in children in the high PEL category.

The results for speed-agility and LLMS can be understood by the association between the improvements in neural pathways produced by the muscle strength gains and movement control and are in line with other PE interventions designed to increase fundamental motor skills [22,24,37,38]. Our intervention used a similar pedagogical approach as the one used in Faigenbaum et al. [22,38] and Sinđić et al. [24] (15 min INT intervention, two times per week for 8 weeks) and Duncan et al. [37] with a 30 to 40 min INT session replacing the regular PE class, once per week for 10 weeks. The participants in the study by Faigenbaum et al. [38] improved the lower and upper body and abdominal strength and the CRF after the intervention. In the study by Duncan et al. [37], participants achieved improvements in muscular strength and velocity in both genders. The study by Faigenbaum et al. [22] found benefits of different magnitude between genders, with females being the ones that improved their results in the abdominal and LLMS and CRF. The study by Sinđić et al. [24] was conducted in females only and found benefits in the ULMS and abdominal strength, but they did not find benefits in CRF. The null effect of our intervention in CRF is in accordance with the study by Sinđić et al. [24] and other studies designed to improve motor skills [39]. In addition, the short duration, and the type of intervention, focused on improving MC instead of on increasing fitness could also explain the inefficacy of the INT program to improve CRF. Some reviews have shown that PE interventions designed to produce changes in fitness are better at increasing fitness compared to PE interventions designed to improve PA or motor skills [3,20]. In addition, the low level of adherence to PA recommendations at baseline could have reduced the efficacy of the intervention, as indicated in a recent review revealing that school-based PA interventions have less effect in CRF in those children with lower levels of vigorous PA at baseline [40].

Referring to BMI, the results are like others [3,20] that showed small changes in BMI, BMI z-score or skinfold thickness or fat measurements. Nevertheless, it is encouraging to report such effects on BMI when the design of the present study was to produce changes in MC, without including any advice or counselling to modify diet habits.

When looking at the effect of the intervention by gender, Faigenbaum et al. [22] found that females had a better response to the INT intervention, and they attributed such an effect to their lower baseline levels of motor skills and fitness compared to males and to gender differences in the neuromuscular system’s capacity to adapt to the intervention. It was found that both boys and girls improved their results in several fitness tests. Although boys in the present study had a better fitness profile than girls, both showed a low baseline fitness level. For all the physical fitness measures except the ULMS, boys had mean values around the 50th percentile for their age, while girls had mean values below that percentile [31,41]. For the ULMS, both boys and girls showed mean values around the 75th percentile for their age [42]. With such low levels of fitness, it is worrisome to look at the results of the children in the schools who did not receive the intervention; after 12 weeks of regular PE class, there were non-significant changes in fitness. Probably, the three-month interval was not sufficient to produce such changes as the effects of regular PE classes on PA and fitness are likely to appear at the end of the school year. Nevertheless, it is encouraging to notice that a brief 20-min INT incorporated into the PE class was sufficient to improve several fitness parameters. According to the literature, girls tend to be less active and less motivated and satisfied, with a negative attitude towards the PE class [43]. If a short intervention such as the one presented in this study could improve the level of perceived physical competency among girls or any children with low level of fitness, it could also improve their participation and enjoyment of the PE class [3,44] and result in higher levels of physical activity outside of school. Schools have autonomy to apply the most appropriate methodology to achieve the objectives of the PE curriculum. Although it is not known what pedagogical model can assure a physically active lifestyle, the INT program explored in this study can be incorporated as one strategy to improve motor competence [23] and physical fitness in children.

Another aim of the study was to assess whether adherence to lifestyle behaviors (PA, MD quality and sleep time) mediates improvements in fitness. As less than 5% of the sample followed the combined lifestyle behaviors, the analysis was limited to the recommendation with the highest level of adherence in the sample; this is physical activity. The results showed that the intervention was not mediated by PA levels at baseline, with non-compliant participants also improving their capacities as measured for speed-agility and muscular strength and the anthropometric measurements. On the contrary, non-compliant children in the control group attending the regular PE class did not obtain benefits in fitness. Regarding other lifestyle behaviors, a concerning result from the present study is the low level of adherence to the MD. Although it could be attributed to the high percentage of non-Caucasian (namely mostly from Subsaharian Africa) children in the study, with a diet pattern distinctive to their country of origin, only a small percentage of children in both groups had good levels of adherence to the MD (24% in the non-Caucasian children and 36% for the Caucasian participants. This aligns with a recent analysis highlighting a concerning decline in diet quality among Spanish children over the past two decades. Between 2000 and 2020, the average KIDMED score for eight- to nine-year-olds dropped from 7.4 to 6.7 [45]. The deterioration of the diet quality can be attributed to several aspects such as the globalization of food production and the widespread adoption of the Western-type diet or the lack of time to dedicate to the home preparation of food [45]. Despite the evidence emphasizing the health benefits of following the MD [46], being sufficiently active and reducing sedentary time [47], having adequate motor competence [10] and being fit [3], there are multiple factors acting as barriers against adopting such habits and skills, such as a low socioeconomic level, lack of time or environmental factors [48], and they should be tackled at early ages to revert such trends.

The last goal was to study if the effects of the intervention differed in children from families with low and high PEL scores. It seems that the intervention was more effective in children from a high PEL, with a higher effect size in speed-agility and ULMS, and for its effect on the BMI. It is possible that the difference in the response to the intervention might be affected by differences in family lifestyle behaviors. Children with highly educated parents dedicate more time to PA [26], have a higher diet quality and show healthier lifestyle clusters [49]. A further analysis was conducted to check if the adherence to the MD and PA recommendations differed according to the parents’ educational level). Although both groups had a medium diet quality, children from families with a low educational level had a significantly lower diet score than children from families with a high PEL. The same was observed for PA; those children whose parents had a higher educational level spent more hours per day being active at moderate to vigorous intensity than children whose parents had a lower level of education.

Thus, it is urgent to revert the trends by adopting policies that involve several actors (schools, families, communities) to develop, implement and evaluate action plans to increase the possibilities for children to be active, to acquire the basic motor skills, to enjoy being active and to become literate on PA and healthy lifestyles. Schools are one setting to develop such plans, but they should be one of many in a broader policy that adopts the so-called strength-based approach to empower the individual to adopt healthier lifestyle behaviors [50].

The present study has some limitations that should be noted. Changes in PA were not assessed throughout the 3-month intervention, although it is unlikely that they could occur after the short duration of the intervention. Second, a follow-up period after the end of the intervention would allow us to evaluate the stability of the changes observed in fitness and analyze changes in out-of-school physical activity. And finally, as the study was conducted in a small region of Spain, the external validity of the research is limited.

## 5. Conclusions

The INT, a short and easy to apply intervention, can be added as a warmup to physical education class to improve physical fitness in school-age children. The program seems especially adequate for children with low levels of physical activity and children with different backgrounds and socioeconomic levels. Its benefits are not limited to motor competence and fitness, but probably extend to the well-known effects of PE such as self-esteem and health.

The adherence to health-related lifestyle behaviors is low in both boys and girls. Schools, together with the participation of families and communities, should be committed to providing young children with the skills to become motivated to adopt healthy lifestyles.

## Figures and Tables

**Figure 1 jfmk-09-00117-f001:**
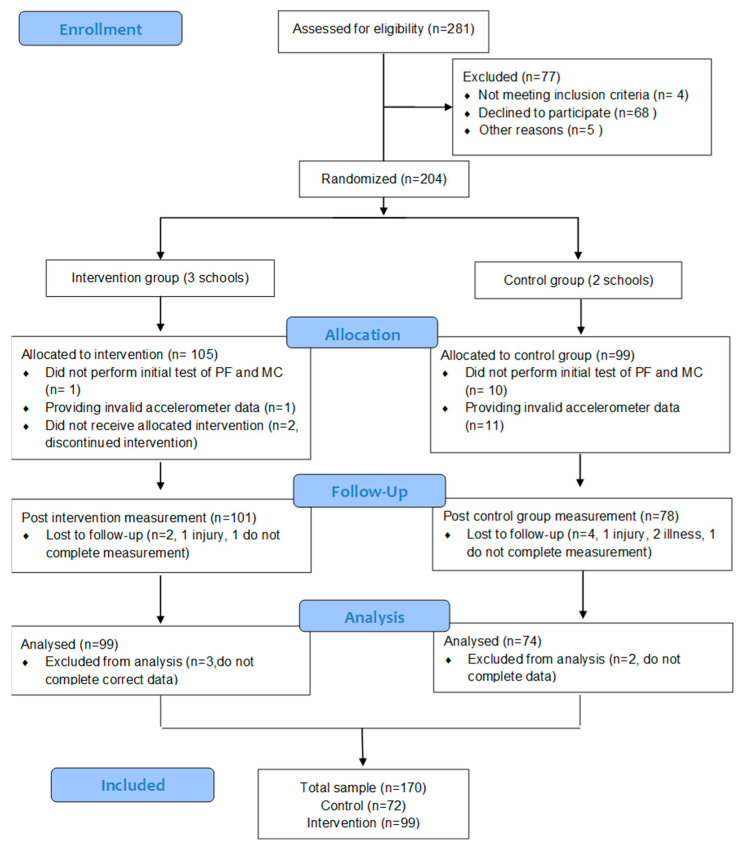
Flowchart of research methodology.

**Table 1 jfmk-09-00117-t001:** Sample characteristics and differences by gender.

	All (*n* = 170)	Boys (*n* = 82)	Girls (*n* = 88)	*p*-Value
Clinical Variables				
Age (y)	7.45 ± 0.34	7.43 ± 0.32	7.47 ± 0.35	0.529
Weight (kg)	25.95 ± 4.2	25.68 ± 3.74	26.23 ± 4.77	0.409
Height (cm)	124.5 ± 14.89	124.7± 15.03	124.4 ± 14.76	0.869
BMI (kg/m^2^)	16.18 ± 2.27	16.08 ± 1.72	16.29 ± 2.83	0.566
BMI SDS	−0.29 ± 0.75	−0.38 ± 0.68	−0.21 ± 0.83	0.154
Fat Mass (%)	20.38 ± 6.07	18.74 ± 4.97	21.72 ± 6.57	0.001
Caucasian (%)	73.9	76.9	71.1	--
High Parental Education (%)	41.63	33.58	40.28	--
Physical Fitness				
Speed-agility (s)	25.02 ± 2.81	23.99 ± 2.58	26.05 ± 3.04	<0.0001
Cardiorespiratory fitness (min)	5.22 ± 0.69	4.93 ± 0.61	5.52 ± 0.78	<0.0001
Lower-limb muscular strength (cm)	96.20 ± 14.37	103.54 ± 16.09	88.87 ± 12.65	<0.0001
Upper-limb muscular strength (kg)	10.5 ± 2.47	10.69 ± 2.37	10.31 ± 2.58	0.348
Physical Activity				
Sedentary (min/d)	450.69 ± 75.69	441.85 ± 81.43	459.54 ± 69.96	0.130
Light (min/d)	293.05 ± 48.13	290.87 ± 56.74	295.24 ± 39.53	0.560
MVPA (min/d)	66.01 ± 21.87	73.69 ± 24.95	58.34 ± 18.79	<0.0001
Lifestyle behaviors				
Mediterranean Diet (points)	6.29 ± 2.10	6.29 ± 1.93	6.29 ± 2.25	0.993
Sleep Time (h)	9.54 ± 1.07	9.35 ± 1.15	9.73 ± 0.99	0.022
Adherence to lifestyle recommendations (%)				
Physical Activity (%)	55.3	70.7	40.9	<0.0001
Mediterranean Diet (%)	31.2	28.0	34.1	0.395
Sleep Time (%)	29.4	23.2	35.2	0.086

BMI: body mass index; SDS: standard deviation score; MVPA: moderate and vigorous physical activity; *p*-values are from an unpaired *t*-test.

**Table 2 jfmk-09-00117-t002:** Effects of the intervention on fitness by gender.

		Boys			Girls		
		Δ (Post-Pre)	*p*-Value	Cohen’s Effect Size	Δ (Post-Pre)	*p*-Value	Cohen’s Effect Size
Speed-agility (s) *	Control	−0.46 ± 1.43	0.111	0.24	−0.28 ± 1.45	0.319	0.15
	Intervention	−1.73 ± 1.11	<0.0001	0.7	−1.40 ± 1.38	<0.0001	0.51
	*p*-value	0.002			0.004		
CRF (min) *	Control	−0.12 ± 0.47	0.241	0.2	−0.039 ± 0.58	0.765	0.07
	Intervention	−0.14 ± 0.56	0.220	0.36	−0.11 ± 0.51	0.245	0.16
	*p*-value	0.87			0.637		
LLMS (cm)	Control	0.99 ± 10.76	0.645	0.06	2.07 ± 8.28	0.189	0.18
	Intervention	5.24 ± 9.35	0.012	0.32	5.79 ± 9.72	0.004	0.51
	*p*-value	0.144			0.124		
ULMS (kg)	Control	0.21 ± 1.10	0.369	0.1	−0.10 ± 1.24	0.723	0.04
	Intervention	0.26 ± 1.05	0.144	0.1	0.76 ± 1.21	<0.0001	0.3
	*p*-value	0.870			0.011		
BMI SDS * (kg/m^2^)	Control	−0.02 ± 0.43	0.452	0.05	−0.03 ± 0.45	0.157	0.05
	Intervention	0.03 ± 0.19	0.211	0.07	−0.11 ± 0.42	<0.0001	0.12
	*p*-value	0.159			0.055		
Fat Mass (%)	Control	−0.02 ± 1.47	0.896	0.01	−0.14 ± 1.73	0.568	0.03
	Intervention	0.35 ± 3.26	0.367	0.07	0.89 ± 2.40	0.001	0.13
	*p*-value	0.506			0.037		

CRF: cardiorespiratory fitness; LLMS: lower-limb muscular strength; ULMS: upper-limb muscular strength; BMI SDS: body mass index standard deviation score. *p*-value from an unpaired *t*-test and the univariate general linear model. * Variables with opposite metric orientation.

**Table 3 jfmk-09-00117-t003:** Effects of the intervention on fitness by physical activity recommendations and parental educational level.

		Physical Activity Recommendations				Parental Education			
		Non-Compliant		Compliant		Low		High	
		Δ (Post-Pre)	Cohen’s Effect Size	Δ (Post-Pre)	Cohen’s Effect Size	Δ (Post-Pre)	Cohen’s Effect Size	Δ (Post-Pre)	Cohen’s Effect Size
Speed-agility (s) *	Cont	−0.42 ± 1.57	0.2	−0.32 ± 1.33	0.16	−0.50 ± 1.56 †	0.021	−0.34 ± 1.51	0.161
	Int	−1.24 ± 1.19 ‡	0.53	−1.84 ± 1.35 ‡	0.73	−1.43 ± 1.64 ‡	0.59	−1.62 ± 1.21 ‡	0.72
	*p*-value	0.039		<0.0001		0.006		<0.001	
CRF (min) *	Cont	−0.13 ± 0.56	0.23	−0.04 ± 0.50	0.07	−0.16 ± 0.52	0.228	−0.06 ± 0.50	0.097
	Int	−0.05 ± 0.51	0.07	−0.20 ± 0.56	0.28	−0.11 ± 0.70	0.14	−0.28 ± 0.63 ‡	0.34
	*p*-value	0.622		0.286		0.741		0.137	
LLMS (cm)	Cont	−0.26 ± 9.05	0.02	2.96 ± 9.67	0.19	2.98 ± 9.41 †	0.187	1.60 ± 11.04	0.097
	Int	5.64 ± 7.67 ‡	0.37	5.44 ± 11.12 †	0.32	5.76 ± 12.54 ‡	0.34	2.62 ± 16.10	0.015
	*p*-value	0.016		0.372		0.229		0.758	
ULMS (kg)	Cont	0.25 ± 1.21	0.11	0.30 ± 1.10	0.14	−0.09 ± 1.28	0.04	0.13 ± 1.18	0.053
	Int	0.61 ± 1.28 ‡	0.27	0.48 ± 1.06 ‡	0.22	0.45 ± 1.49	0.17	1.04 ± 1.25 ‡	0.4
	*p*-value	0.023		0.525		0.125		0.003	
BMI SDS *(kg/m^2^)	Cont	−0.05 ± 0.19	0.07	0.01 ± 0.16	0.04	−0.03 ± 0.27	0.01	−0.07 ± 0.15 ‡	0.37
	Int	−0.69 ± 0.19 †	0.09	0.01 ± 0.29	0.04	−0.05 ± 0.21 †	0.01	−0.05 ± 0.20	0.44
	*p*-value	0.764		0.928		0.554		0.527	
Fat Mass (%)	Cont	−0.10 ± 1.77	0.01	0.01 ± 1.44	0.01	0.28 ± 1.50	0.05	−0.30 ± 1.60	0.05
	Int	−0.85 ± 2.72 †	0.14	−0.59 ± 3.04	0.09	−0.83 ± 3.49 †	0.12	−0.51 ± 1.77 †	0.08
	*p*-value	0.114		0.208		0.012		0.561	

CRF: cardiorespiratory fitness; LLMS: lower-limb muscular strength; ULMS: upper-limb muscular strength; BMI SDS: body mass index standard deviation score. *p*-value from unpaired *p*-value from unpaired *t*-test and the univariate general lineal model. * Variables with opposite metric orientation. † *p* < 0.05; ‡ *p* < 0.001.

**Table 4 jfmk-09-00117-t004:** Logistic regression model examining physical fitness (≥percentile 50) according to the intervention, by compliance with physical activity (PA) recommendations and parental education level (PEL).

	All ChildrenM OR (95% CI)	Non-Compliant (PA) OR (95% CI)	Compliant (PA) OR (95% CI)	Low PEL OR (95% CI)	High PEL OR (95% CI)
Speed-agility					
Basal	2.51 (1.84–3.42) ***	3.03 (1.77–5.20) ***	2.48 (1.54–3.99) ***	3.22 (1.91–5.43) ***	2.22 (1.39–3.53) ***
Intervention	2.50 (0.92–6.79)	2.93 (0.61–14.10)	1.60 (0.36–7.11)	2.90 (0.70–11.99)	1.92 (0.37–9.96)
Cardiorespiratory fitness					
Basal	6.16 (3.06–12.40) ***	8.62 (2.71–27.37) ***	6.02 (1.94–18.66) ***	5.91 (2.72–16.04) ***	7.64 (2.36–24.79) **
Intervention	0.81 (0.37–1.80)	0.46 (0.13–1.55)	1.27 (0.37–4.36)	0.51 (0.16–1.58)	1.94 (0.52–7.29)
Lower-limb muscular strength					
Basal	1.14 (1.09–1.18) ***	1.19 (1.07–1.32) ***	1.13 (1.06–1.21) ***	1.17 (1.10–1.26) ***	1.09 (1.03–1.14) **
Intervention	1.19 (0.54–2.62)	0.61 (0.12–3.10)	4.64 (1.10–19.54) *	1.23 (0.39–3.85)	1.28 (0.38–4.26)
Upper-limb muscular strength					
Basal	1.67 (1.45–1.92) ***	1.87 (1.41–2.48) ***	1.65 (1.35–2.01) ***	2.87 (1.86–4.42) ***	4.40 (2.21–8.79) ***
Intervention	0.62 (0.25–1.53)	0.63 (0.13–3.09)	0.38 (0.10–1.43)	1.16 (0.30–4.49)	0.08 (0.01–0.60) *
Body mass index					
Basal	24.19 (8.89–65.82) ***	13.89 (3.99–48.35) ***	175 (11.00–2684) ***	12.81 (4.34 –37.82) ***	86.44 (8.31–899.06) ***
Intervention	0.18 (0.05–0.58) **	0.14 (0.03–0.80) *	0.20 (0.03–1.20)	0.52 (0.14–1.96)	0.07 (0.00–1.00) *
Fat mass (%)					
Basal	2.66 (2.03–3.48) ***	2.70 (1.75–4.18) ***	3.14 (2.02–4.89) ***	9.42 (4.17–21.28) ***	9.38 (3.41–25.80) ***
Intervention	1.28 (0.48–3.41)	4.75 (0.83–27.27)	1.93 (0.40–9.44)	2.70 (0.67–10.88)	0.91 (0.22–3.82)

* *p* < 0.02; ** *p* < 0.001; *** *p* < 0.0001; Adjusted for gender and BMI.

## Data Availability

Dataset available on request from the authors.

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
