# Peer review of "Lifestyle as a Modulator of the Effects on Fitness of an Integrated Neuromuscular Training in Primary Education"

_jfmk, 2024, doi:10.3390/jfmk9030117_

Round 1
Reviewer 1 Report
Comments and Suggestions for Authors
I would like to thank for the opportunity to review this manuscript. Overall, this manuscript is written on an important topic, and it reads well. However, there is also room for the improvement.
Introduction of this manuscript is very short. Please elaborate the introduction of this manuscript. Please provide more detailed explanation of important concepts related to this research. For example, Authors only briefly mention the term motivation, but they do not introduce any prominent motivational theory. For example, the self-determination theory is a prominent motivational theory because it differs from different forms of motivation. Also please add more information why physical fitness is important in early ages and how physical education classes can contribute to physical fitness. Why physical education classes are important in relation to physical activity? Authors haven’t provided any evidence on this. For example, based on the transcontextual model of motivation, motivation towards physical activity in PE classes tend to transform into motivation towards physical activity during leisure time, which has also been shown in studies where physical activity was measured via both questionnaires and accelerometers (see Kalajas-Tilga et al., 2022).
Kalajas-Tilga, H., Hein, V., Koka, A., Tilga, H., Raudsepp, L., & Hagger, M. S. (2022). Trans-Contextual Model Predicting Change in Out-of-School Physical Activity: A One-Year Longitudinal Study. European Physical Education Review, 28(2), 463–481. https://doi.org/10.1177/1356336X211053807
Subjects of this study could be described in a more detail.
Please provide more information about measurements and structure it based on the measured variable.
Please provide more information about the intervention program. Currently, description about intervention program is very short.
Figure 1 is hard to read, please increase the quality of Figure 1.
Not sure if tables meet journal guidelines, please check.
Table 3 is messy, please revise.
Table 4 is with low quality, please revise.
For intervention studies, you need to statistically conduct randomization check and attrition check. To test the intervention effects, you need to conduct repeated measures ANCOVA. Authors need to rerun their statistical analysis. Based on this, I cannot comment the discussion, because the discussion needs to be rewritten when new statistical analysis are conducted.
Authors hardly mention any limitations of this study. Please add more limitations as well as suggestions for future research based on these limitations.
Conclusions is too short, please elaborate.
Reviewer 2 Report
Comments and Suggestions for Authors
I would like to thank authors for this interesting study, it is very well organized and there is a merit for this journal.
The purpose of this study was to determine the changes on fitness after an integrated neuro- 25 muscular training (INT) intervention in primary school children and to assess how lifestyle behaviors and parental education modulate those changes.
One hundred and seventy children were participants. Cardiorespiratory fitness (½ mile run test), 10 × 5 m shuttle run test, standing broad jump (SBJ), handgrip dynamometer, body mass index (BMI) and fat mass percentage (FM%) were assessed before and after the 3 months intervention (20 minutes of INT in the physical education class, twice per week). In addition, Mediterranean Diet (MD), sleep time and parental education level (PEL) were assessed by questionnaires and adherence to physical activity (PA) recommendations was measured using a triaxial accelerometer before the intervention.
Results indicated that there were improvements in the 10x5 test and the SBJ. Only girls had improvements in the handgrip test, BMI SDS and FM%. After correcting for confounding variables only BMI was significantly improved whereas strength improved in the participants non-compliant with the PA recommendations or pertaining to families of high PEL.
In summary, the INT produced improvements in fitness in a brief period and in different subgroups of pupils such as inactive and with diverse sociocultural environments.
I believe this is an interesting study but here is my review and feedback: Authors did such a nice job in terms of design of this study. However, they should report more clear the validity and reliability of tests and measurements in this study. In addition, more up to date literature review need (2020 and above) to improve literature review and discussion parts. Moreover, a practical applications should be recommended for schools and future research in detail. Besides, if they can explain what is Mediterranean diet in detail and its measurement, this would be very helpful.
Finally, I liked that authors used a control group and after study over they did intervention as well. However, my question is about experimental group. Did authors follow up children physical activity and life styles after at home and after school to make it more clear? Because children may do different activities after school and this may influence the result. I would like to than authors for this interesting study and hard work. I look forward to seeing edited version of this valuable manuscript. Best regards.
Reviewer 3 Report
Comments and Suggestions for Authors
Although the introduction is rich in citations, it lacks a comprehensive review and critical analysis of the literature. For example, the relationship between PA, MC, and fitness is merely presented through scholars’ opinions or research results. This section should be improved and expanded to include more critical analysis and synthesis of the literature. The current approach mainly reports results from other articles, with minimal discussion and comparison of the actual findings with existing literature.
The research sample is limited to a specific region, which lacks broad representativeness. It is recommended to explicitly state the research limitations.
I recommend highlighting the exceptional reliability and validity of the research tool, specifically the triaxial Actigraph GT3X accelerometer (Actigraph, Pensacola, FL, USA).
In the discussion section, the paper mainly reports results from other articles. The actual findings are poorly discussed and not sufficiently compared with the current literature. Furthermore, the results are not critically analyzed, and no practical applications are proposed.
I recommend adding one or two more specific conclusions to highlight the main results of the study.
Reviewer 4 Report
Comments and Suggestions for Authors
L67: avoid using etc. If etc is important, then list all things associated with it. If not use (e.g., )
L110: what qualifications did the INT expert have
Table 3: formatting to get the p-values to be in one line vs. wrapping to the next line
In the discussion you use 1st person language (we, our); tried to use third person language
In the limitations section you discuss “some limitations”, however, I only see one reported
Author Response
Please see the attachement

Round 2
Reviewer 1 Report
Comments and Suggestions for Authors
Authors have done well job on revising their manuscript.
Reviewer 2 Report
Comments and Suggestions for Authors
I believe that current version of this manuscript is acceptable in the present form. Authors answered all questions. It is very well done. Best regards.